# Structural and mechanistic basis of proton-coupled metal ion transport in the SLC11/NRAMP family

Ines A. Ehrnstorfer[1,*], Cristina Manatschal[1,*], Fabian M. Arnold[1,†], Juerg Laederach[1,†] & Raimund Dutzler[1]

Secondary active transporters of the SLC11/NRAMP family catalyse the uptake of iron and manganese into cells. These proteins are highly conserved across all kingdoms of life and thus likely share a common transport mechanism. Here we describe the structural and functional properties of the prokaryotic SLC11 transporter EcoDMT. Its crystal structure reveals a previously unknown outward-facing state of the protein family. In proteoliposomes EcoDMT mediates proton-coupled uptake of manganese at low micromolar concentrations. Mutants of residues in the transition-metal ion-binding site severely affect transport, whereas a mutation of a conserved histidine located near this site results in metal ion transport that appears uncoupled to proton transport. Combined with previous results, our study defines the conformational changes underlying transition-metal ion transport in the SLC11 family and it provides molecular insight to its coupling to protons.

[1] Department of Biochemistry, University of Zurich, Winterthurerstrasse 190, 8057 Zurich, Switzerland. * These authors contributed equally to this work. † Present addresses: Institute of Medical Microbiology, Gloriastrasse 30/32, University of Zurich, 8006 Zurich, Switzerland (F.M.A.); Institute of Molecular Biology and Biophysics, ETH Zurich, Otto Stern-Weg 5, 8093 Zurich, Switzerland (J.L.). Correspondence and requests for materials should be addressed to R.D. (email: dutzler@bioc.uzh.ch).

The transition metals $Fe^{2+}$ and $Mn^{2+}$ are essential trace elements in all kingdoms of life. Their transport across cellular membranes is catalysed by members of the solute carrier 11 (SLC11) or natural resistance-associated macrophage protein (NRAMP) family[1,2]. These proteins are highly conserved and function as secondary active transporters[3]. Whereas family members poorly discriminate between different divalent transition-metal ions, they efficiently select against $Ca^{2+}$ and $Mg^{2+}$, which are both several orders of magnitude more abundant[4,5]. With respect to function, the best-characterized transporter is human SLC11A2 (or divalent metal ion transporter 1, DMT1)[4,6]. This protein is widely expressed in different tissues[4,7]. In enterocytes, it is located at the apical side of the epithelium where it mediates the uptake of $Fe^{2+}$ (refs 4,8). In all other cells, DMT1 is found in intracellular membranes where it promotes the exit of endocytosed $Fe^{2+}$ from endosomes into the cytoplasm[9]. DMT1 was shown to catalyse the cotransport of $Fe^{2+}$ and $H^+$ but also uncoupled leaks of either substrate have been observed[4,10]. Highly homologous prokaryotic SLC11 family members transport $Mn^{2+}$ into the cytoplasm[11]. Also for prokaryotic homologues, transport was proposed to be coupled to $H^+$ (refs 11–15). Our previous studies on the structure and function of the transition-metal ion transporter from *Staphylococcus capitis* (ScaDMT) have revealed the general architecture of the SLC11/NRAMP family and they provided initial insight into the structural basis of selective transition-metal ion transport[16]. As predicted from sequence analysis[17], the overall fold of ScaDMT resembles distantly related transport proteins that include the amino acid transporter LeuT[18]. The structure showed an inward-facing conformation of the transporter with a site that is accessible from the cytoplasm and that binds transition-metal ions but not calcium[16]. Functional assays with protein reconstituted into liposomes underlined the structural results by demonstrating that ScaDMT transports various transition-metal but not alkaline earth-metal ions[16]. Despite the advance in the understanding of ion selectivity, important properties of the transport mechanism remained elusive. The most important open questions concern the structure of the transporter in an outward-facing conformation and whether, similar to their eukaryotic counterparts, transport in the prokaryotic transporters would be coupled to $H^+$. To resolve these questions we have investigated the structural and functional properties of the SLC11 transporter from the bacterium *Eremococcus coleocola* (EcoDMT). We have optimized our transport assays to show that EcoDMT transports $Mn^{2+}$ with a $K_M$ in the low μM range and that metal ion transport is coupled to the cotransport of protons. By determining the structure of EcoDMT in an outward-facing conformation, we provide a framework that defines the conformational changes underlying transport by the alternate access mechanism[19]. The structure also reveals the location of a histidine that, on mutation, prevents $H^+$ but not $Mn^{2+}$ transport. In conjunction with previous studies, our data provide a solid basis for the comprehension of proton-coupled transition-metal ion transport in the SLC11 family.

## Results

**Functional properties of EcoDMT.** We have identified EcoDMT from a previously described broad screen of prokaryotic homologues[16] as protein with outstanding biochemical properties. EcoDMT is 511 amino acids long and shares a sequence identity of 53% and similarity of 67% with ScaDMT, the prokaryotic family member of known structure (Supplementary Fig. 1a). Compared with this transporter, EcoDMT contains an extension at the C-terminus that encodes for an additional 12th

transmembrane helix, which is also found in all eukaryotic family members (Supplementary Fig. 1a,b). Similar to ScaDMT, its sequence identity of 30% with human DMT1 is remarkably high, which underlines the strong degree of conservation in this family (Supplementary Fig. 1a). EcoDMT was expressed in *E. coli* and purified in the detergent *n*-decyl-β-D-maltopyranoside (DM). Like ScaDMT, the protein is monomeric in solution (Supplementary Fig. 2a). When reconstituted into liposomes and assayed with the help of the metal sensitive fluorophore calcein, trapped inside the proteoliposomes, the protein mediates transport of $Mn^{2+}$ with considerably faster kinetics than ScaDMT. We thus assumed that EcoDMT would be an ideal candidate to overcome the limitations of the assay used for ScaDMT that did not allow for a quantitative characterization of its transport properties. We reasoned that the previous limitations were due to the intrinsic leakiness of the proteoliposomes for protons. This was likely caused by the high protein to lipid ratio, the choice of lipids used during reconstitution, and the use of comparably high ion concentrations required due to the slow kinetics of transport. To overcome these limitations we investigated the reconstitution of EcoDMT with different lipids at lower protein to lipid ratio[20] and we found conditions where we detected little activity in the absence of ionophores and a strong enhancement of $Mn^{2+}$ transport, on addition of valinomycin, which dissipates the membrane potential established by the electrogenic transporter (Supplementary Fig. 2b). This assay allowed us to observe transport already at low μM concentrations of $Mn^{2+}$ with a dose-dependent increase of the activity that levels off at higher concentrations, as expected for a transporter containing a substrate-binding site (Fig. 1a, Supplementary Fig. 2c). When assayed at pH 7.2, metal ion transport saturates with a $K_M$ of about 18 μM (Fig. 1b), which is in the same range as the value obtained for human DMT1 by electrophysiology[4] and for *E. coli* MntH using a cellular uptake assay[15]. The transport activity is $Na^+$ independent (Supplementary Fig. 2d) but dependent on the proton concentration. It accelerates upon a decrease of the outside pH from 7.2 to 6.2 and slows down upon an increase to pH 8.2 (Fig. 1c). We were next interested whether we would see $Mn^{2+}$ dependent uptake of $H^+$ into proteo-liposomes containing EcoDMT, which would be indicative for $H^+$-coupled $Mn^{2+}$ cotransport. We thus monitored the pH decrease within the liposomes with the pH sensitive fluorophore 9-Amino-6-Chloro-2-Methoxyacridine (ACMA) that accumu-lates at the inside of the acidified vesicles[21]. Figure 1d shows the time-dependent quenching of ACMA caused by the transport of $H^+$ accompanying the influx of $Mn^{2+}$ into the vesicles. The quenching is dependent on the gradient of the transition-metal ion with higher concentrations on the outside showing faster kinetics. Only a shallow decay is observed in the absence of $Mn^{2+}$, despite of a driving force for $H^+$ set by the negative membrane potential. To demonstrate that the effect is due to the accumulation of $H^+$ and not the interaction of $Mn^{2+}$ with ACMA, we have monitored the same transport at high internal buffer concentration, which prevents a change of pH. Under this condition, we did not see any decrease in fluorescence (Supplementary Fig. 2e). Contrary, the presence of intravesicular EDTA, which scavenges free $Mn^{2+}$, enhances the fluorescence change (Supplementary Fig. 2e). In our acidification experiments, the apparent $Mn^{2+}$ dependence is moderately shifted towards higher concentrations compared with experiments where we detect metal ion uptake (Fig. 1a,d). This can be expected due to the presence of buffer inside the liposomes, which decreases the sensitivity of the assay. Our data thus strongly suggest that EcoDMT functions as secondary active transporter that couples the transport of $Mn^{2+}$ to the transport of $H^+$ in the same direction, with a $K_M$ in the low μM range.

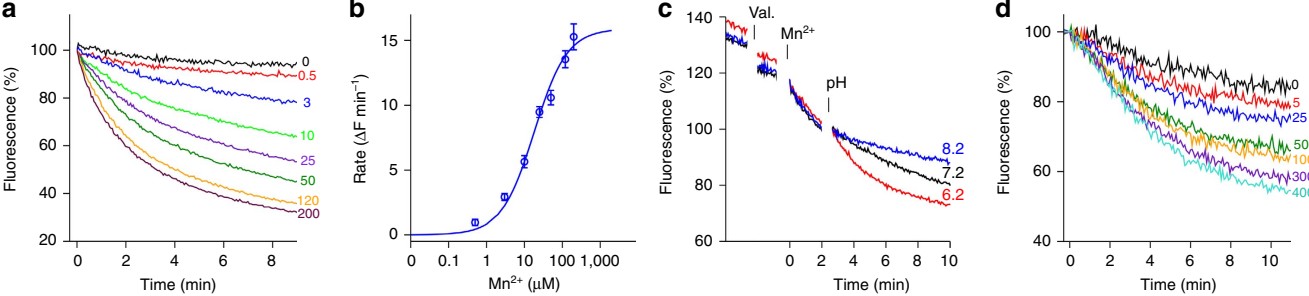

**Figure 1 | Functional characterization of EcoDMT.** (**a**) Metal ion transport. EcoDMT mediated $Mn^{2+}$ transport into proteoliposomes assayed by the quenching of the fluorophore calcein trapped inside the vesicles. Experiments were carried out at pH 7.2. Traces are shown in unique colours with outside $Mn^{2+}$ concentrations (µM) indicated. (**b**) $Mn^{2+}$ concentration dependence of transport. The solid line shows the fit to a Michaelis–Menten equation with an apparent $K_M$ of 18.2 µM. Data points represent mean and s.e.m. of 5–9 technical replicates from two independently prepared batches of proteoliposomes. (**c**) pH dependence of transport. Time dependence of $Mn^{2+}$ transport at an outside concentration of 4.5 µM. Addition of valinomycin (Val.) and $Mn^{2+}$ are indicated. Transport was initially assayed at pH 7.2 for all conditions. After two minutes (pH) the pH was adjusted to the indicated value by addition of equivalent volumes of either NaOH, water, or HCl. (**d**) Proton transport. $Mn^{2+}$-dependent transport of $H^+$ into proteoliposomes containing EcoDMT. Transport is assayed by the quenching of the pH-dependent fluorophore ACMA at an initially symmetric pH of 7.2. Traces are shown in unique colours with outside $Mn^{2+}$ concentrations (µM) indicated.

**Structure of EcoDMT in an outward facing conformation**. To gain novel insight into the structural properties of SLC11 family members we have crystallized EcoDMT and determined its structure by X-ray crystallography at a resolution of 3.3 Å (Table 1). The crystals are of space group C2 and contain one copy of the molecule in the asymmetric unit. All attempts to use the inward-facing conformation of ScaDMT for phasing by molecular replacement were unsuccessful, which suggested that, in this crystal form, EcoDMT exhibits substantial structural differences. We thus grew crystals of a selenomethionine derivatized protein and obtained a set of initial phases from a three wavelength multiple anomalous dispersion experiment. After their improvement by solvent flattening, these phases allowed for model building and refinement (Table 1, Supplementary Fig. 3a–c). The correct register of the structure was confirmed by the anomalous difference density of selenium atoms defining the location of 14 native methionines (excluding two disordered positions at the N-terminus) and seven additional positions obtained from data collected from several point mutants where methionines were introduced in regions that lack this residue in the native protein (Supplementary Fig. 3d,e). Most of the structure, including the entire membrane-inserted part, is well defined in the electron density except for 13 residues on the N- and 5 residues on the C-terminus. The structure of EcoDMT is shown in Fig. 2a. As anticipated from the sequence, the protein comprises 12 transmembrane helices. The terminal helix (α-helix 12) of EcoDMT, which is absent in ScaDMT, is located at the periphery of the protein and does probably not play a major role during transport (Fig. 2a). Like ScaDMT[16] and other transporters sharing a similar fold[18], EcoDMT displays a pseudo-symmetric relationship between two structurally analogous domains consisting of α-helices 1–5 and α-helices 6–10 (Fig. 2a, Supplementary Fig. 1b). Both domains are intertwined and aligned in opposite orientation within the membrane. Helices 1 and 6 are both unwound in the center of the lipid bilayer and, as revealed in the ScaDMT structure[16], conserved residues of this unwound region constitute the transition-metal ion-binding site (Fig. 2b). In this site, ion–protein interactions are established with a backbone carbonyl oxygen at the end of α-helix 6a, and the side chains of an aspartate and an asparagine, all of which act as hard ligands that would also be suitable for the coordination of $Ca^{2+}$. In contrast, the side chain of a methionine that is also part of the binding site acts as transition metal specific soft ligand that would not be suitable for $Ca^{2+}$ coordination[16,22] (Supplementary

Fig. 4a). The high sequence similarity to ScaDMT allows for a reliable comparison of equivalent segments. Both structures delineate different conformations on the transport cycle with ScaDMT adopting an inward-facing and EcoDMT an outward-facing state (Fig. 3a–c, Supplementary Fig. 4b–d). 305 equivalent Cα positions from 11 transmembrane helices superimpose with a root mean square deviation (RMSD) of 3.3 Å but structural differences are unevenly distributed and the average RMSD of single transmembrane α-helices ranges from 1.5 to 6.0 Å (Fig. 3b). Compared with ScaDMT there are only moderate differences in the main-chain positions of α-helices 2, 3, 6b, 7, 8, 9 and the N-terminal half of α-helix 4, whereas the conformation has changed considerably for α-helices 1b and 10 and even more extensively for α-helices 5, 6a and the C-terminal half of α-helix 4. Large conformational changes are also expected for α-helix 1a, which, although well defined in EcoDMT, cannot be quantified due to its absence in the ScaDMT structure. The largest deviation between the two structures concerns the C-terminal half of α-helix 4 and α-helix 5 both of which have moved in a nearly concerted manner by a hinge-like rotation around an axis running from the center of α-helix 4 to the end of α-helix 5 (Fig. 3d). In the transition from the outward- to the inward-facing conformation, this movement opens a gap between α-helices 5 and 7 on the intracellular side, which is filled by the movement of α-helix 1a on opening of an intracellular access route to the ion-binding site (Fig. 3c,d). Conversely, the aqueous path to the extracellular side is closed by a large movement of α-helix 6a and smaller movements of α-helices 1b, 10 and the extracellular part of α-helix 2 (Fig. 3c,e). As consequence of the described conformational changes, the access to the ion-binding site from the cytoplasm found in ScaDMT is closed in the EcoDMT structure whereas the residues constituting the ion-binding site are now accessible from the outside via an aqueous cavity with a diameter between 5 and 9 Å (Fig. 3c, Supplementary Fig. 4b–d).

**Functional properties of ion-binding site mutants**. Although in the structure of EcoDMT, the residues constituting the transition-metal ion-binding site are exposed to the extracellular side, we did not observe anomalous difference density indicative for bound $Mn^{2+}$ or other transition-metal ions in soaks or co-crystallization experiments. This is in contrast to the expected high affinity of ions binding to an outward-facing conformation and may be related to the high pH of our crystallization solutions and

**Table 1 | Data collection and refinement statistics**

| | EcoDMT | EcoDMT Se-Met | | |
|---|---|---|---|---|
| *Data collection* | | | | |
| Space group | C2 | C2 | | |
| Cell dimensions | | | | |
| a, b, c (Å) | 149.2, 81.7, 96.2 | 148.4, 80.9, 96.1 | | |
| α, β, γ (°) | 90, 107.6, 90 | 90, 107.3, 90 | | |
| | | *Peak* | *Inflection* | *Remote* |
| Wavelength (Å) | 1.0 | 0.9794 | 0.9796 | 0.9173 |
| Resolution (Å) | 50–3.3 (3.4–3.3)* | 50–3.6 (3.7–3.6) | 50–3.6 (3.7–3.6) | 50–3.6 (3.7–3.6) |
| $R_{merge}$ (%) | 4.9 (157) | 6.4 (109.2) | 6.1 (125.2) | 6.3 (144.4) |
| $CC_{1/2}$ (%) | 99.8 (87.4) | 99.9 (85.5) | 99.9 (79.9) | 99.9 (73.0) |
| $I/\sigma I$ | 26.6 (1.8) | 12.9 (1.7) | 13.3 (1.5) | 12.9 (1.3) |
| Completeness (%) | 99.7 (99.5) | 99.8 (99.9) | 99.8 (100.0) | 99.8 (99.9) |
| Redundancy | 13.6 (14.3) | 7.1 (7.4) | 7.1 (7.3) | 7.1 (7.4) |
| | | | | |
| *Refinement* | | | | |
| Resolution (Å) | 12–3.3 | | | |
| No. Reflections | 16,390 | | | |
| $R_{work}/R_{free}$ (%) | 23.2/27.7 | | | |
| No. atoms | | | | |
| Protein | 3,780 | | | |
| Ligand/ion | 33 | | | |
| Water | — | | | |
| *B* factors | | | | |
| Protein | 181.6 | | | |
| Ligand/ion | 179.8 | | | |
| r.m.s. deviations | | | | |
| Bond lengths (Å) | 0.002 | | | |
| Bond angles (°) | 0.52 | | | |

*Values in parentheses are for highest-resolution shell.

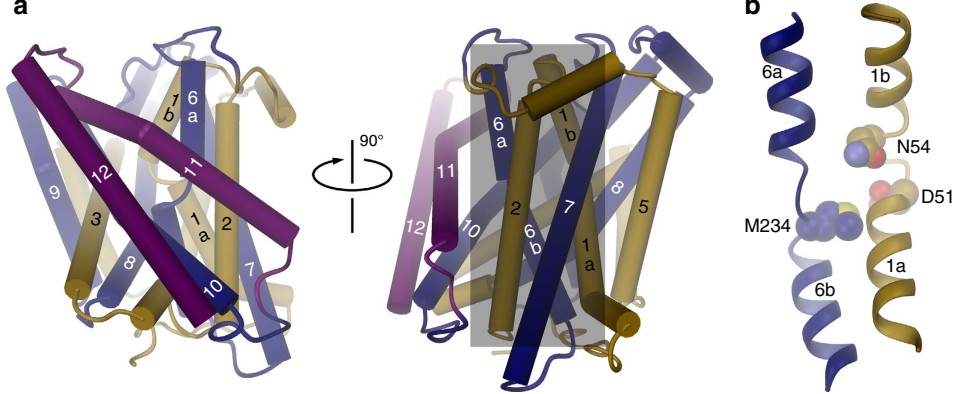

**Figure 2 | EcoDMT Structure.** (**a**) Presentation of EcoDMT in two different orientations. Helices are displayed as cylinders. The perspective is from within the membrane. The N-terminal sub-domain (encompassing α-helices 1–5) is coloured in beige, the C-terminal sub-domain (encompassing α-helices 6–12) in dark blue, α-helices 11 and 12 in magenta. The grey box in the right panel indicates the region viewed in **b**. (**b**) Ribbon representation of the interrupted α-helices 1 and 6. The view is approximately as in a (right panel). Side chains of residues constituting the ion-binding site are shown as space-filling models. Molecular representations in Figs 2–5 were prepared with DINO (http://www.dino3d.org/).

the impediment of substrate-induced conformational changes on metal ion binding by the crystalline environment. Compared with the ScaDMT structure, in the EcoDMT structure, the access to the ion-binding site is considerably wider and the distance between metal-coordinating residues is larger (Fig. 4a, Supplementary Fig. 4e). Particularly, the interaction of the ion with the backbone of the C-terminus of α-helix 6a that is established in the ScaDMT structure may only be formed on the transition of the protein into a substrate-occluded conformation (Fig. 4a, Supplementary Fig. 4e).

To confirm that the same residues identified in ScaDMT and human DMT1 (ref. 16) also account for ion recognition in EcoDMT, we have studied the transport behaviour of binding-site mutants in our *in vitro* assay. For that purpose, we have purified and reconstituted the mutants D51A, N54A and M234A in which the side chains of the corresponding residues that were found to interact with transition-metal ions in ScaDMT[16] were truncated to alanine (Supplementary Fig. 4f). In all three mutants we have observed a severely perturbed transport behaviour with strongly reduced activity, indicative for a decreased affinity for the substrate (Fig. 4b). Among the binding-site mutants the effect appears least severe in M234A, for which we have detected residual transport at high $Mn^{2+}$ concentrations. In a next step, we were interested whether a protein with compromised

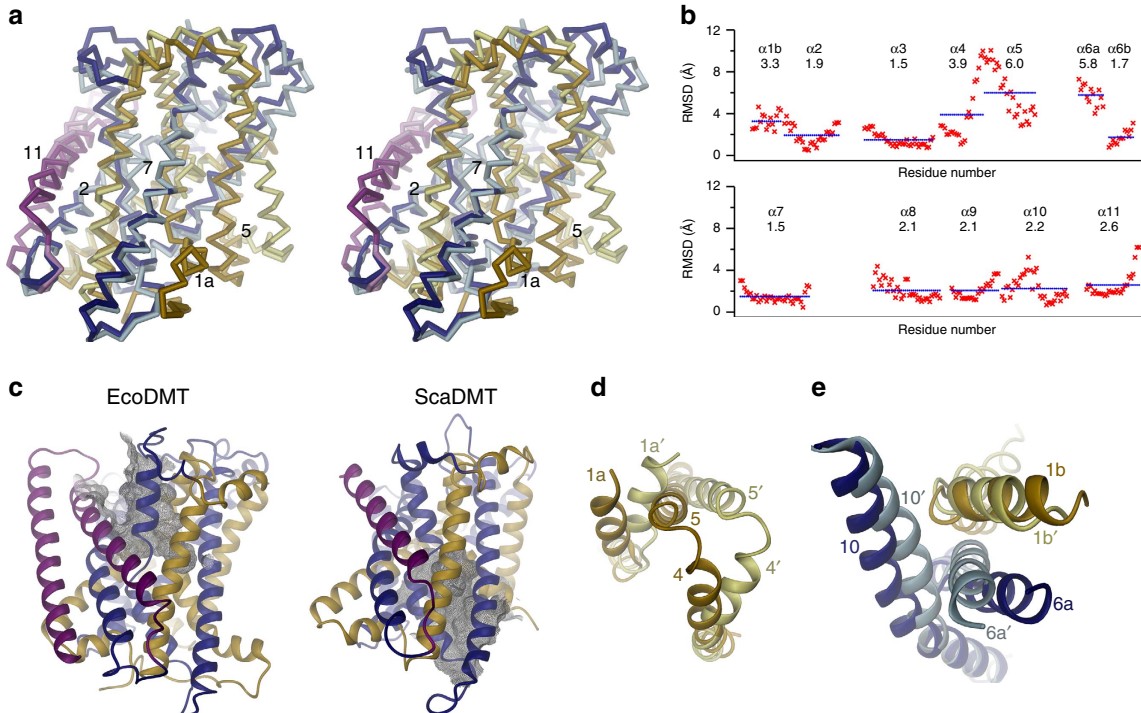

**Figure 3 | Comparison of EcoDMT and ScaDMT.** (**a**) Stereo view of a superposition of EcoDMT and ScaDMT (PDB ID 5M94). The proteins are shown as Cα-traces. Perspective and colour-coding of EcoDMT are as in Fig. 2a (right panel). For ScaDMT, the N-terminal domain is coloured in yellow, the C-terminal domain in light blue and α-helix 11 in pink. (**b**) RMSD of Cα positions calculated from a least-square superposition of equivalent regions of ScaDMT and EcoDMT. The residue number is plotted on the x-axis. Blue lines and numbers show averages for each transmembrane segment. (**c**) Aqueous cavities leading to the metal binding sites of EcoDMT (left) and ScaDMT (right). Proteins are coloured as in Fig. 2 and shown as ribbons. The view is approximately as in the left panel of Fig. 2a. Parts of the molecular surface showing the access cavities are shown as dense grey mesh. Movements in an outward to inward transition regulating the opening of the intracellular cavity (**d**) and the closing of the extracellular cavity (**e**). In **d** the view is from the intra- and in **e** from the extracellular side along an axis perpendicular to the membrane. Parts of the EcoDMT and ScaDMT structures are shown as ribbons and coloured as in **a**. The position of α-helix 1a in the ScaDMT structure was modelled on an intermediate position based on the corresponding region in the inward-facing structures of LeuT (PDB ID 3TT3) and vSGLT (PDB ID 3DH4) (Supplementary Fig. 4c).

ion-binding properties would still show proton transport. We thus characterized $H^+$ influx into proteoliposomes containing the EcoDMT mutants at concentrations up to $400\,\mu M\;Mn^{2+}$, conditions at which we detected low activity in M234A and barely any $Mn^{2+}$ transport in D51A and N54A. In no case, we did observe acidification of the lumen of the vesicles (Fig. 4c), which is expected if $H^+$ transport in EcoDMT is coupled to the transport of $Mn^{2+}$. It thus appears that the transport of protons requires conformational changes of the ion-binding site and that, by mutation of this site, we did not create an uncoupled proton leak of the transporter.

**Mutations of residues involved in proton transport.** Since the transport of protons appears to be coupled to conformational changes of the protein, we attempted to pinpoint residues in EcoDMT that may serve as $H^+$ acceptors for this process. We thus analysed the outward-facing EcoDMT and the inward-facing ScaDMT structure with respect to residues that could be protonated and that have changed their accessibility to either side of the membrane. In this analysis, we identified two candidates, Glu129 and His236, that are both located in vicinity of the binding site and that are highly conserved within the SLC11 family (Fig. 5, Supplementary Fig. 1a). To investigate the role of these residues for function, we have prepared the point mutants E129Q, E129A and H236A, determined their structure by X-ray crystallography and investigated their $Mn^{2+}$ and $H^+$ transport properties with our fluorescence-based *in vitro* assays. The crystal

structures of all three mutants are very close to wildtype (WT), confirming that the respective mutation did not interfere with folding (Supplementary Fig. 5, Table 2). When assaying the transport of $Mn^{2+}$ into proteoliposomes, all three mutants show a dose-dependent quenching of the fluorescence of calcein trapped inside the vesicles, thus demonstrating that they still transport transition-metal ions (Fig. 6a). A kinetic analysis reveals that the $Mn^{2+}$ concentration dependence of all three mutants is similar to WT, while the maximum activity is moderately decreased in both mutants of Glu129 and reduced to half in H236A. (Fig. 6b, Supplementary Fig. 5d). As for WT, $Mn^{2+}$ transport by E129Q is accelerated at lower pH and attenuated at higher pH (Fig. 6c). Conversely, no pH dependent change in the transport activity was observed for E129A and H236A (Fig. 6c). When assaying the uptake of $H^+$, E129Q shows a WT-like $Mn^{2+}$ dependent change in the intravesicular pH, whereas no pH change is detected for H236A even at high $Mn^{2+}$ concentrations (Fig. 6d,e). In contrast to E129Q and H236A, the mutation E129A resulted in a strong $H^+$ leak conductance even in the absence of $Mn^{2+}$ (Fig. 6d,e). Acidification is independent of metal ions and proceeds with similar kinetics as the maximal activity observed for E129Q at high $Mn^{2+}$ concentrations (Fig. 6d,e).

Collectively, our results on the characterization of mutants of two conserved ionizable residues close to the substrate-binding site have revealed their different role in proton-coupled metal ion transport. Whereas the mutation of a glutamate to alanine has induced a strong uncoupled $H^+$ leak even in the absence of metal

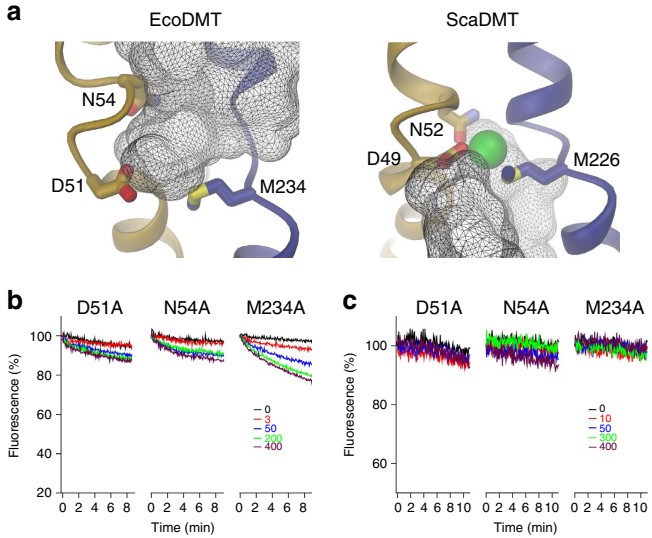

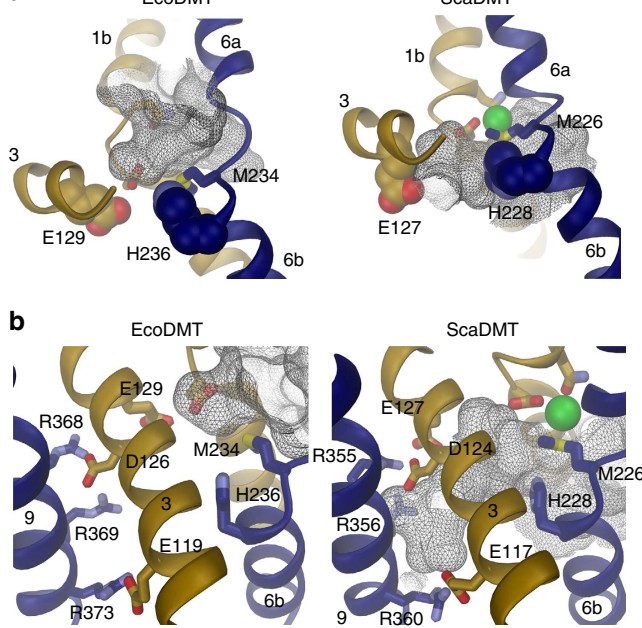

**Figure 4 | Properties of ion-binding site mutants.** (**a**) Access to the ion-binding site in the outward-facing conformation of EcoDMT (left) and the inward-facing conformation of ScaDMT (right). The view is rotated by about 180° compared with Fig. 2b. Parts of α-helices 1 and 6 are shown as ribbon, the side chains of ion coordinating residues as sticks. The molecular surface of aqueous cavities leading to the ion-binding site is shown as grey mesh. The crystallographically defined $Mn^{2+}$ ion in ScaDMT is depicted as green sphere. (**b**) $Mn^{2+}$ transport into proteoliposomes containing EcoDMT mutants D51A (left), N54A (center) and M234A (right). Transport is assayed by the quenching of the fluorophore calcein trapped inside the vesicles. Experiments were carried out at pH 7.2. (**c**) $Mn^{2+}$-dependent transport of $H^+$ into proteoliposomes containing the EcoDMT mutants D51A (left), N54A (center) and M234A (right). Transport is assayed by the quenching of the pH-dependent fluorophore ACMA at an initially symmetric pH of 7.2. Traces in **b** and **c** are shown in unique colours with outside $Mn^{2+}$ concentrations (μM) indicated.

**Figure 5 | Residues potentially involved in $H^+$ transport.** (**a**) Location of potential proton acceptors with respect to the ion-binding site of EcoDMT (left) and ScaDMT (right). The side chains of ion coordinating residues are shown as sticks, Glu129 and His236, two potential proton acceptors, as space-filling models. (**b**) Location of Glu129 and His236 with respect to the aqueous substrate access cavities in EcoDMT (left) and ScaDMT (right). Side chains of ion coordinating residues, the two potential proton acceptors and of conserved acidic and basic residues lining a narrow aqueous cavity in ScaDMT are shown as sticks. In **a** and **b**, the molecular surface of aqueous cavities is shown as grey mesh. The green sphere corresponds to the bound $Mn^{2+}$ in ScaDMT. Selected α-helices are shown as ribbons and labelled accordingly.

ions, the conservative mutation of the same residue to glutamine, thereby removing its ability to accept and release a proton, had little influence on the functional behaviour, thus indicating that this residue does not function as primary $H^+$-acceptor. In contrast, the mutation of a conserved histidine on α-helix 6b to alanine disrupts the coupling of $Mn^{2+}$ to $H^+$ without creating uncoupled leaks, which implies that this residue likely plays a central role in $H^+$ transport.

## Discussion

In the present study, we have characterized the structural and functional properties of the prokaryotic SLC11 transporter EcoDMT. Combined with our previous work, its crystal structure has provided a framework that defines the conformational transitions during transport of transition-metal ions by the alternate-access mechanism[19,23–25] (Fig. 7a, Supplementary Movie 1). The previously determined ScaDMT structure has disclosed an inward-facing conformation with a substrate-binding site that is accessible from the intracellular side[16]. The structure of the outward-facing state of EcoDMT has now revealed the opposite endpoint within the transport cycle. Taken together, both structures permit the comparison with other transporters sharing a similar fold. By now, the structures of several such transporters for amino acids[18,26–30], neurotransmitters[31,32], sugars[33] and other compounds[34–38] have been determined. For the sake of clarity, the following analysis focuses on a comparison with the amino acid transporter LeuT, for which multiple conformations are known[18,26] (Supplementary

Fig. 6a,b) and which is one of the closest, structural homologs of the SLC11 family[16]. With less than 14% of identical residues (determined in a pairwise alignment), there is no obvious sequence relationship between the two proteins. The RMSD between equivalent Cα atoms (encompassing 241 positions and excluding α-helix 12, which is in a different conformation) of EcoDMT and the outward-facing conformation of LeuT (PDB ID 5JAE), is 3.5 Å (Supplementary Fig. 6c). The corresponding value to the inward-facing conformation (PDB ID 3TT3) of 5.1 Å clearly underlines the structural equivalence of EcoDMT to an outward-facing state. Conversely, 224 Cα positions of the inward-facing conformations of ScaDMT and LeuT superimpose with an RMSD of 3.9 Å (Supplementary Fig. 6d). The comparably high RMSD values between the corresponding states of the two SLC11 transporters and LeuT emphasize significant structural differences between the two protein families. Nevertheless, both transporters undergo similar conformational changes (Figs 3a and 7a, Supplementary Fig. 6a). This is illustrated in an analysis of the RMSDs of single positions between the two states of each transporter (Fig. 3b, Supplementary Fig. 6b). In both cases, certain regions of the molecule undergo large movements whereas others are essentially invariant. In LeuT the invariant parts of the structure were termed scaffold domain and they were assigned to α-helices 3, 4, 8 and 9, whereas α-helices 1, 2, 5, 6 and 7 constitute the moving core domain[26]. In general, a similar pattern is also found for SLC11 transporters, although there are differences when comparing the extent of movements. Compared with LeuT, the conformational changes at the C-terminal part of α-helix 4

**Table 2 | Data collection and refinement statistics of EcoDMT mutants.**

|  | EcoDMT E129Q | EcoDMT E129A | EcoDMT H236A |
|---|---|---|---|
| *Data collection* |  |  |  |
| Space group | C2 | C2 | C2 |
| Cell dimensions |  |  |  |
| $a, b, c$ (Å) | 151.3, 81.8, 96.5 | 149.1, 80.8, 96.4 | 150.4, 81.8, 96.4 |
| $\alpha, \beta, \gamma$ (°) | 90.0, 107.6, 90.0 | 90.0, 107.4, 90.0 | 90.0, 107.5, 90.0 |
| Resolution (Å) | 50–3.6 (3.7–3.6)* | 50–3.9 (4.0–3.9) | 50–3.7 (3.8–3.7) |
| $R_{merge}$ (%) | 6.3 (154) | 5.9 (112.6) | 5.6 (140.9) |
| $CC_{\frac{1}{2}}$ (%) | 99.8 (81.7) | 99.7 (79.0) | 99.9 (87.0) |
| $I/\sigma I$ | 12.8 (1.8) | 13.4 (1.8) | 17.2 (2.0) |
| Completeness (%) | 98.7 (99.8) | 98.4 (98.3) | 98.9 (99.9) |
| Redundancy | 6.7 (7.0) | 6.8 (6.6) | 11.2 (11.4) |
|  |  |  |  |
| *Refinement* |  |  |  |
| Resolution (Å) | 12–3.6 | 12–3.9 | 12–3.7 |
| No. reflections | 12658 | 9644 | 11546 |
| $R_{work}/R_{free}$ (%) | 22.1/26.5 | 21.1/26.2 | 23.2/26.8 |
| No. atoms |  |  |  |
| Protein | 3,780 | 3,776 | 3,775 |
| Ligand/ion | — | — | — |
| B factors |  |  |  |
| Protein | 213.4 | 211.2 | 201.3 |
| Ligand/ion | — | — | — |
| r.m.s. deviations |  |  |  |
| Bond lengths (Å) | 0.002 | 0.002 | 0.002 |
| Bond angles (°) | 0.49 | 0.49 | 0.50 |

*Values in parentheses are for highest-resolution shell.

and α-helix 5, which, in conjunction with the movement of α-helix 1a controls the access to the intracellular binding site, appear to be significantly larger in SLC11 transporters (Fig. 3a,b, Supplementary Fig. 6a,b). This movement involves a kink in the center of α-helix 4 that is not observed in LeuT (Fig. 3a,d, Supplementary Fig. 6a). Similarly, the movement of α-helix 6a, which closes the extracellular access to the binding site, is larger compared with LeuT, whereas the conformational change of α-helix 1b is smaller (Fig. 3b,e, Supplementary Fig. 6b). These structural differences have affected the shape of the aqueous pockets leading to the binding site of SLC11 transporters. Whereas the access to the substrate-binding site in LeuT in the respective state is granted via a single wide funnel-shaped aqueous path, its equivalence in SLC11 transporters is branched and narrower in one direction, in line with the smaller size of the transported substrate (Supplementary Fig. 7a).

Unlike most transporters sharing the same fold that function as Na$^+$-coupled symporters[18,32–34,36,39], transport in the SLC11 family is driven by the cotransport of H$^+$, with a presumable stoichiometry of one proton per divalent metal ion[4]. The coupling to H$^+$ provides a source of energy for active transport in such different environments as the plasma membrane and membranes of intracellular organelles. Despite of their abundance in pro- and eukaryotic organisms, the mechanisms of proton-coupled transporters are to date still poorly understood. In our study, we have demonstrated that metal ion transport in EcoDMT is coupled to H$^+$ (Fig. 1d). We observed H$^+$ transport even at pH 7.2 where uncoupled transport of divalent transition-metal ions was proposed for hDMT1 (ref. 10). Since our assays did not allow us to investigate H$^+$ transport at lower pH, we cannot exclude the possibility of uncoupled leaks at higher proton concentrations. To gain deeper insight into the transport mechanism, we were interested in pinpointing residues that would be involved in H$^+$-coupling. Our mutagenesis experiments involving residues of the metal ion-binding site showed that acidification is not detectable in mutants where the decreased affinity for the substrate prevented efficient transport, and we thus concluded that H$^+$ transport is connected to a change in the access of the substrate-binding region. We took this observation into account when selecting residues that may act as proton acceptors during transport. No suitable candidates were found in regions corresponding to two Na$^+$-binding sites found in LeuT[18] or other related transporters[33,36] (Supplementary Fig. 7a,b). Similarly, the region around a basic amino acid that was proposed to play a role in proton transport in the presumably H$^+$-coupled transporter ApcT[27] did not contain any residues that may accept and release a proton (Supplementary Fig. 7d). In contrast, we found two protonatable residues, Glu129 and His236, that are located close to the binding site and that change their access to either side of the membrane from the outward- to the inward-facing conformation (Fig. 5). Both residues are highly conserved within the family (Supplementary Fig. 1a). Whereas the pKa of a histidine side chain is in a suitable range to accept and release a H$^+$ at slightly acidic pH, the pKa of Glu129 would have to be shifted upwards. In the outward-facing conformation, Glu129 is located close to the surface of the aqueous substrate cavity in vicinity to the metal ion coordinating Asp51 (Fig. 5a,b). We suspected that the electrostatic interaction with the negatively charged binding-site residue might stabilize its protonated state and in turn position Asp51 to interact with the transition-metal ion. Yet, we did not detect an altered behaviour in transport experiments of the conservative mutant E129Q, since the mutant transports Mn$^{2+}$ and H$^+$ with similar properties as WT (Fig. 6a–d). This suggests that Glu129 is not the primary H$^+$ acceptor during transport. Still, since at lower pH, the corresponding mutation of the *E. coli* homologue MntH strongly decreased the rate of Mn$^{2+}$ uptake into cells[40], this residue probably plays an important role for protein function.

In contrast to E129Q, the mutation of His236 displays a pronounced altered phenotype, which could be explained by a role in accepting and releasing a proton during transport. The mutant H236A shows robust Mn$^{2+}$ uptake but no detectable

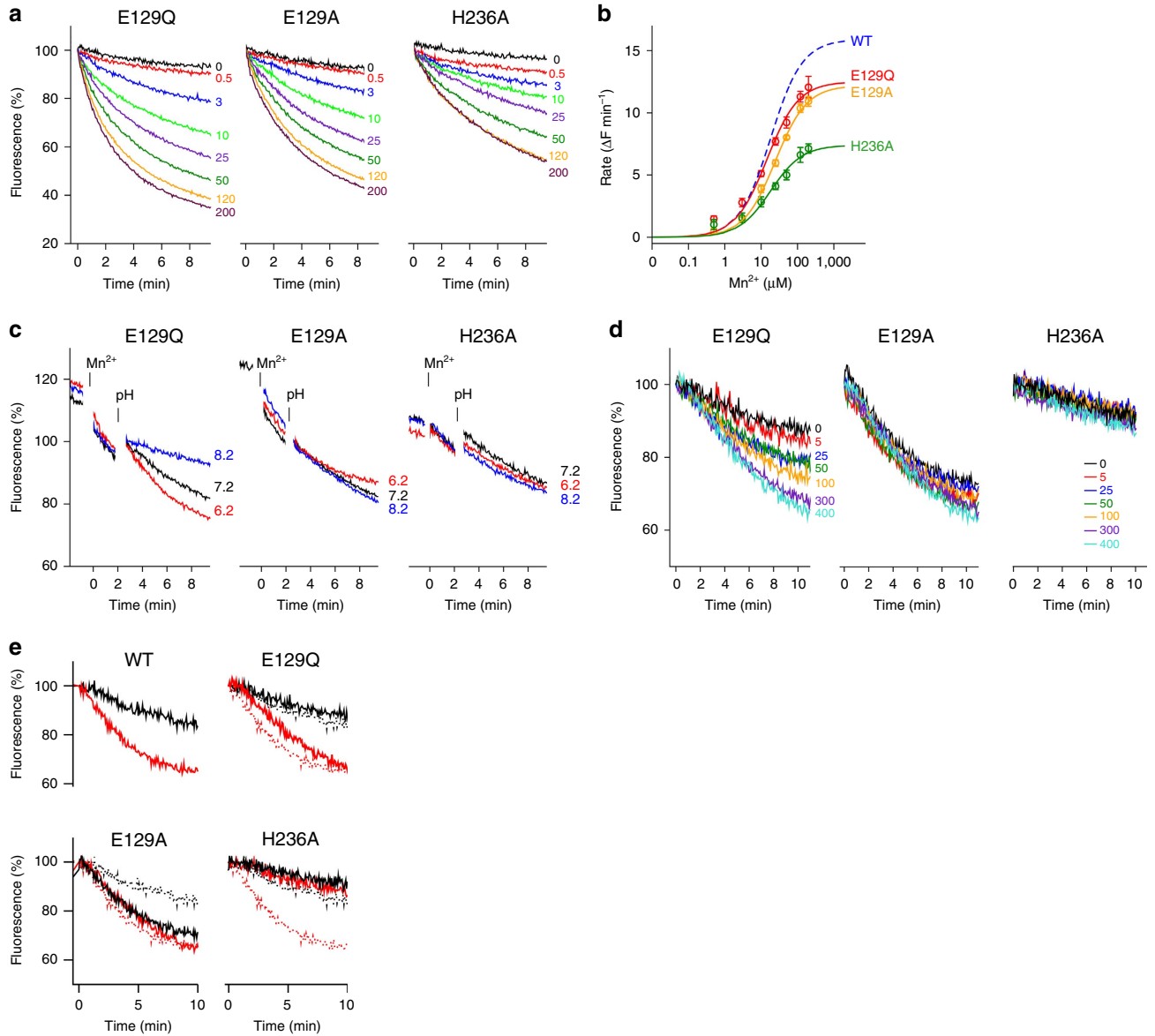

**Figure 6 | Functional properties of mutants of potential H$^+$ acceptors.** (**a**) Mn$^{2+}$ transport into proteoliposomes containing the EcoDMT mutants E129Q (left), E129A (center) and H236A (right). Experiments were carried out at pH 7.2. Traces are shown in unique colours with outside Mn$^{2+}$ concentrations (μM) indicated. (**b**) Mn$^{2+}$ concentration dependence of transport in the mutants E129Q, E129A and H236A. The solid lines show fits to a Michaelis-Menten equation (E129Q, $K_M$ 14.4 μM, $v_{max}$ 12.5 ΔF min$^{-1}$; E129A, $K_M$ 24.4 μM, $v_{max}$ 12.2 ΔF min$^{-1}$; H236A, $K_M$ 18.1 μM, $v_{max}$ 7.4 ΔF min$^{-1}$). WT ($K_M$ 18.2 μM, $v_{max}$ 15.9 ΔF min$^{-1}$) is shown for comparison. Data points represent mean and s.e.m. of 4–7, 5–10 and 3–6 technical replicates for E129Q, E129A and H236A, respectively. (**c**) pH dependence of transport into proteoliposomes containing mutants E129Q (left), E129A (center) and H236A (right). The experiment was carried out as described for Fig. 1c. (**d**) Mn$^{2+}$-dependent transport of H$^+$ into proteoliposomes containing the mutants E129Q (left), E129A (center) and H236A (right). Traces are shown in unique colours with outside Mn$^{2+}$ concentrations (μM) indicated. (**e**) Comparison of the acidification of proteoliposomes at similar Mn$^{2+}$ transport rates. Traces in the absence of Mn$^{2+}$ are shown in black. In case of mutants, red traces correspond to a Mn$^{2+}$ concentration of 300 μM, and in case of WT to 100 μM to account for its higher maximal activity. Data are from Figs 1d and 6d. Mutant panels show WT traces as dotted line for comparison.

coupling to H$^+$ (Fig. 6a,d). However, due to its lower maximal activity and the comparably low sensitivity of our H$^+$ transport assay, we cannot exclude the possibility of residual H$^+$ transport that may have escaped our detection. Remarkably, His236 was previously assigned a potential role in proton coupling and pH dependence of transport in human DMT1 (ref. 41), whereas smaller effects on mutation of this residue were observed in other studies of the same protein[10] and for MntH[14,15,42].

In the outward-facing structure of EcoDMT, His236 is located close to the surface of the aqueous path leading to the transition-metal ion-binding site and in the inward-facing structure of

ScaDMT it is placed at the crossroad of two potential intracellular proton exit pathways (Fig. 5a,b). In the latter conformation, His236 is on one side exposed to the wide inwardly-directed metal ion release path of the protein. Along this path, it is in proximity of His241, a second conserved histidine placed at α-helix 6b, which was implicated to play a role in transport in different family members[10]. In the other direction, it faces a narrow, presumably water-filled pocket that has become accessible due to the movement of Glu129 away from the metal ion-binding site (Fig. 5b). This pocket is lined by interacting acidic residues located on α-helix 3 (in EcoDMT Glu119, Asp126,

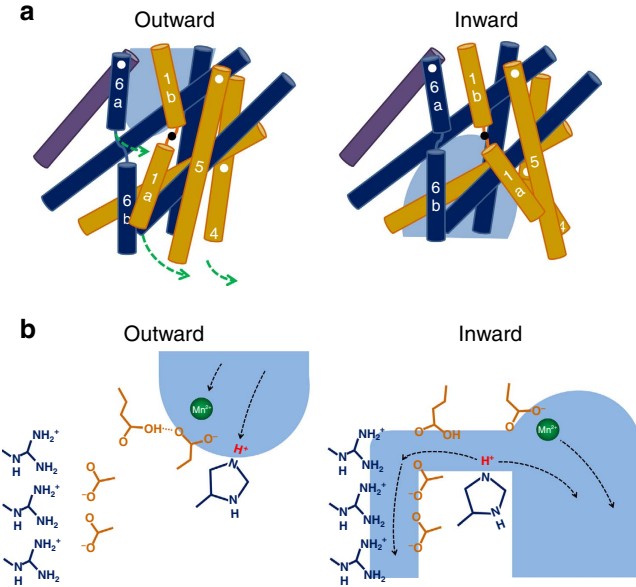

**Figure 7 | Transport mechanisms. (a)** Schematic depiction of the alternate access mechanism in SLC11 transporters. Selected helices are shown, and moving parts are labelled. Hinges and direction of movements are indicated as circles and arrows. Aqueous access paths are shown in light blue. **(b)** Potential mechanism of $H^+$-coupled $Mn^{2+}$ cotransport. For the metal ion-binding site only the coordinating aspartate is depicted. In the outward facing state (left) the conserved histidine acts as proton acceptor. After the transition to an inward-facing state (right), the $Mn^{2+}$ ion exits via the main aqueous path. The proton may either be released via the same path as $Mn^{2+}$ or via a narrow aqueous cavity that is lined by conserved acidic and basic residues. Uncoupled $H^+$ transport in human DMT1 and in the EcoDMT mutant E129A may also take place via this narrow aqueous cavity.

Glu129) and basic residues on α-helix 9 (Arg368, Arg369, Arg373) (Fig. 5b). Most of these residues (except Arg369) are conserved between family members, whereas the equivalent region in LeuT and other $Na^+$-coupled transporters is predominantly hydrophobic (Supplementary Fig. 7e). Interestingly, Arg416 in human DMT1, which corresponds to Arg373 in EcoDMT, was found to be mutated in a patient with severe anemia[43,44]. Since the described ionizable residues bridge His236 to the intracellular surface of the molecule, they are potential constituents of an $H^+$ exit path (Fig. 5b). The same region may also conduct uncoupled leak in the absence of $Mn^{2+}$ that was observed in human DMT1 (ref. 10) and in the EcoDMT mutant E129A, where the removal of the side chain may have opened a proton conduction path that does no longer require a change of conformation of the transporter.

We thus end up with two possible scenarios for the $H^+$-coupled transition-metal ion transport in the SLC11 family that are illustrated in Fig. 7b. In both cases His236 acts as primary acceptor for the proton in an outward-facing state. After transition to the inward-facing state, the proton is released either via the main metal ion exit path or a smaller side tunnel that both connect to the cytoplasm. Due to the strong conservation, we assume that this mechanism is shared by other members of the SLC11 family. Despite the strong evidence for the role of His236 as acceptor during proton transport, there are further open questions. It is currently unclear how the protonation of His236 would facilitate the binding of divalent transition-metal ions or vice versa and how the protein transits between opposite states, particularly since the positively charged side chain would have to cross the membrane without stabilization by a close-by negative countercharge. In summary, our studies on EcoDMT have

deepened our understanding of transition-metal ion transport in the SLC11 family and they have provided important insight into the coupling to protons. A full comprehension of this process will require extended further investigations for which we have here provided a structural as well as a functional foundation.

## Methods

**Protein expression and purification.** The gene coding for the divalent metal ion transporter of the bacterium *Eremococcus coleocola* (EcoDMT; UniProtKB identifier E4KPW4) was amplified from its genomic DNA and cloned into the L-arabinose-inducible expression vector pBXC3GH with fragment-exchange (FX) cloning[45]. Point mutations were introduced by site-directed mutagenesis (Supplementary Table 1). For expression *E. coli* MC1061 (ref. 46) carrying the pBXC3GH-EcoDMT plasmid were grown by fermentation in Terrific Broth (TB) medium supplemented with 0.75% (v/v) glycerol and 100 μg ml$^{-1}$ ampicillin. The temperature was gradually decreased from 37 to 25 °C. At an $OD_{600}$ of about 2.5, protein expression was induced by addition of 0.0045% (w/v) L-arabinose and proceeded overnight at 18 °C. All following steps were performed at 4 °C. After harvest by centrifugation, cells were lysed in buffer A (50 mM potassium phosphate pH 7.5, and 150 mM NaCl) containing 1 mg ml$^{-1}$ (w/v) lysozyme and 20 μg ml$^{-1}$ DNaseI. The lysate was cleared by centrifugation (10,000 g for 20 min) and the membranes were harvested by ultracentrifugation (200,000 g for 1 h). Membranes were resuspended in buffer A containing 10% (w/v) glycerol and proteins were extracted for 1 h by addition of 1.5% (w/v) *n*-decyl-β-D-maltopyranoside (DM, Anatrace). Insoluble parts were removed by ultracentrifugation and the solubilized protein was purified by immobilized metal affinity chromatography (IMAC). The GFP-His$_{10}$ tag was removed by incubation with HRV 3C protease for 2 h. The imidazole concentration was lowered by dialysis against 20 mM HEPES pH 7.5, 150 mM NaCl, 8.7% (w/v) glycerol, and 0.25% (w/v) DM during cleavage. Histidine-tagged GFP and the protease were subsequently removed by IMAC. The cleaved protein was concentrated and applied to a Superdex S200 size exclusion chromatography column (GE Healthcare) equilibrated in 10 mM HEPES pH 7.5, 150 mM NaCl, and 0.25% (w/v) DM. The peak fractions were concentrated and used for subsequent crystallization experiments and reconstitution into liposomes. WT and mutants were purified using the same procedures. The molecular weight of EcoDMT was determined by Multi-angle light scattering at 20 °C on an HPLC (Agilent 1100) connected to an Eclipse 3 system equipped with a miniDAWN TREOS MALS detector and an Optilab T-rEX refractometer (Wyatt Technology). For that purpose 40 μg of purified WT (at 0.8 mg ml$^{-1}$) was injected onto a Superdex S200 column (GE Healthcare) equilibrated in 10 mM HEPES pH 7.5, 150 mM NaCl, and 0.25% (w/v) DM. Data was analysed with the Astra package (Astra 6.1, Wyatt Technology).

**Expression of selenomethionine labelled protein.** For preparation of selenomethionine labelled protein[16,47], *E. coli* MC1061 cells carrying the pBXC3GH-EcoDMT plasmid grown in TB medium were diluted 1:100 in M9 medium supplemented with trace elements, Kao and Michayluk Vitamin solution (Sigma), 20 μg ml$^{-1}$ (w/v) thiamine, 0.75% (v/v) glycerol and 100 μg ml$^{-1}$ ampicillin. Growth was initiated at 37 °C and temperature was gradually decreased to 25 °C during 5 h of shaking. To inhibit the methionine synthesis pathway[47], the amino acids L-lysine, L-threonine and L-phenylalanine (each at a concentration of 125 mg l$^{-1}$) and L-leucine, L-isoleucine and L-valine (each at a concentration of 62.5 mg l$^{-1}$) were added and the culture was incubated for one hour before addition of 50 mg l$^{-1}$ of L-selenomethionine. One hour later, expression was induced by addition of 0.0045% (w/v) L-arabinose and proceeded overnight at a temperature of 18 °C. Cells were harvested, lysed and extraction was started from cleared lysate by addition of 1.5% DM at 4 °C. All subsequent purification steps were carried out as described for the non-derivatized protein. The quantitative replacement of methionines by selenomethionine was confirmed by mass spectrometry.

**Crystallization.** Crystals of EcoDMT WT and mutants were obtained in sitting drops at 4 °C using 24-well plates and a reservoir solution containing 50 mM Tris-HCl pH 8.0–9.0 and 22–26% PEG 400 (v/v). Crystallization experiments were prepared by mixing of 1 μl protein solution (at a concentration of 7–10 mg ml$^{-1}$) and 1 μl reservoir solution and drops were subsequently equilibrated against 500 μl reservoir solution. For cryoprotection, the PEG 400 concentration was increased stepwise to 35% (v/v).

**Structure determination.** All data sets were collected on frozen crystals on the X06SA or the X06DA beamline at the Swiss Light Source of the Paul Scherrer Institut on an EIGER X 16M or a PILATUS 6M detector (Dectris). The data was indexed, integrated and scaled with XDS[48] and further processed with CCP4 programs[49]. Phases were obtained from multiple-wavelength anomalous dispersion data collected for selenomethionine derivatized EcoDMT crystals (Table 1). The selenium sites were identified with SHELX C and D[50,51]. Selenium sites were refined in SHARP[52], and phases were improved by solvent flattening with the program DM[53]. In the resulting electron density (Supplementary Fig. 3a), the

position of all transmembrane helices could be identified using the ScaDMT structure (PDB ID 5M94) as a template and the selenium sites as constraints. The model was built in COOT[54] and initially refined in Refmac5 using jelly-body restraints[55]. In later stages of model building, the structure was refined in Phenix[56]. Experimental phase restraints were included to improve convergence. $R_{work}$ and $R_{free}$ were monitored throughout. $R_{free}$ was calculated by selecting 5% of the reflection data that were omitted in refinement. The final model has $R_{work}$ and $R_{free}$ values of 23.2 and 27.7%, good geometry, with 95% of residues in the most favored and none in disallowed regions of the Ramachandran plot (Table 1). The polypeptide chain from residue 13–506 could be traced and almost all side chains could be positioned accurately (Supplementary Fig. 3b,c). The correct register of the amino acid sequence at the N-terminus and between $\alpha$-helices 4 and 6 was confirmed in the anomalous difference density obtained from data of seleno-methionine derivatized crystals of mutants I19M, E27M, W32M, L36M, L154M, V180M and A195M (Supplementary Fig. 3d,e). Crystals of mutants E129Q, E129A, and H236A were refined in Phenix[56] using a modified structure of EcoDMT as starting model (Supplementary Fig. 5). Due to the high sequence homology, the EcoDMT structure has allowed us to correct local errors in the ScaDMT structure. These concern the conformation of the loops connecting $\alpha$-helix 5 with $\alpha$-helix 6a and $\alpha$-helix 6b with $\alpha$-helix 7 and the register of the terminal $\alpha$-helix 11 (Supplementary Fig. 8). The corrections do not affect any conclusions drawn in the previous study[16]. The corrected coordinates of ScaDMT were re-refined and deposited with the PDB under accession codes (5M94 and 5M95).

**Preparation of proteoliposomes.** Proteoliposomes containing EcoDMT were prepared with destabilized liposomes[57]. Synthetic phospholipids dissolved in chloroform (POPE, POPG from Avanti Polar Lipids) at a w/w ratio of 3:1 were used as lipid source[20]. Lipids were dried, washed with diethylether and dried by exsiccation. Lipids were subsequently resuspended and sonicated in buffer containing 20 mM HEPES pH 7.2, and 100 mM KCl. Large unilamellar vesicles were formed by three freeze-thaw cycles followed by extrusion through a 400 nm polycarbonate filter (Avestin, LiposoFast-Basic). The diluted liposomes ($4$ mg ml$^{-1}$) were destabilized by addition of Triton X-100. The purified protein (in DM) was incubated with destabilized liposomes at a protein to lipid ratio of 1:100 (w/w). Detergent was removed by addition of Bio-Beads SM-2 (Bio-Rad) over a period of three days. Proteoliposomes were harvested by centrifugation, resuspended in buffer containing 10 mM HEPES pH 7.2, and 100 mM KCl and stored in liquid nitrogen. All transport experiments were carried out with proteoliposomes reconstituted at a protein to lipid ratio of 1:100 (w/w).

**Fluorescence-based Mn$^{2+}$ transport assay.** For Mn$^{2+}$ transport assays, proteoliposomes were resuspended in buffer B containing 20 mM HEPES pH 7.2, 100 mM KCl and 250 µM calcein (Invitrogen). Samples were flash-frozen in liquid nitrogen and thawed at 25 °C three times before extrusion through a 400 nm polycarbonate filter (Avestin, LiposoFast-Basic). Proteoliposomes were harvested by centrifugation and washed twice by resuspension in 20 volumes of buffer B without Calcein. Control liposomes devoid of protein were prepared in the same way (Supplementary Fig. 2c). For measurement, the sample was diluted to 0.25 mg lipid ml$^{-1}$ in buffer containing 20 mM HEPES pH 7.2 and 100 mM NaCl. For each Mn$^{2+}$ concentration, a 100 µl aliquot was placed in a black 96-well plate. Transport was initialized by addition of MnCl$_2$ and valinomycin (final concentration 100 nM, Sigma-Aldrich) and fluorescence was recorded in 4 s intervals in a fluorimeter (Tecan Infinite M1000; excitation at 492 nm, emission at 518 nm). After 10 min Mn$^{2+}$ ions were equilibrated by addition of the Mn$^{2+}$-H$^+$ exchanger calcimycin (final concentration 100 nM, Invitrogen). A pH gradient was established by addition of HCl or NaOH to the outside solution to reach the desired proton concentration. Initial transport rates ($\Delta F$ min$^{-1}$) were calculated by performing a linear regression on the transport data between 30 and 90 s after addition of valinomycin and MnCl$_2$. The resulting data was fitted to a Michaelis-Menten equation. Kinetic data of WT and all mutants described in this study was confirmed with at least two independent reconstitutions.

**Fluorescence-based H$^+$ transport assay.** For H$^+$ transport assays, proteoliposomes were mixed with buffer containing 6 mM HEPES pH 7.2, 100 mM KCl and 50 µM ACMA (Invitrogen) unless otherwise stated (Supplementary Fig. 2e). The turbid suspension was sonicated until it clarified. Control liposomes devoid of protein were prepared using the same procedure (Supplementary Fig. 2e). For measurement, the sample was diluted to 0.15 mg lipid ml$^{-1}$ in buffer containing 6 mM HEPES pH 7.2, 100 mM NaCl (resulting in a final ACMA concentration of 0.5 µM). For each Mn$^{2+}$ concentration, a 100 µl aliquot was placed in a black 96 well plate. Transport was initialized by addition of MnCl$_2$ and valinomycin (final concentration 4 nM, SigmaAldrich) and fluorescence was recorded in 4 s intervals in a fluorimeter (Tecan Infinite M1000; excitation at 412 nm, emission at 482 nm).

**Data availability.** Coordinates and structure factors for EcoDMT and the mutants E129Q, E129A and H236A have been deposited in the Protein Data Bank with the accession codes 5M87, 5M8K, 5M8A and 5M8J. The corrected coordinates of ScaDMT and the ScaDMT-Mn$^{2+}$ complex have been deposited in the Protein Data Bank with the accession codes 5M94 and 5M95. All the remaining data

supporting the findings of this study can be obtained from the corresponding author on reasonable request. Sequences of EcoDMT, ScaDMT and human DMT1 (UniprotKB identifiers E4KPW4, A0A178L6Y2-1, P49281-2) and PDB accession codes PDB ID 3TT3 (LeuT, inward facing), 5JAE (LeuT, outward facing), 3F3E (LeuT, outward occluded), 3GI8 (APCT), 3DH4 (vSGLT), 4WGV (ScaDMT) and 4WGW (ScaDMT-Mn$^{2+}$ complex) were used in this study.

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

## Acknowledgements

This research was supported by the Swiss National Science Foundation through the National Centre of Competence in Research TransCure. We thank the staff of the X06SA beamline for support during data collection, B. Blattmann and C. Stutz-Ducommun of the Protein Crystallization Center UZH, for their support with crystallization, M.-F. Tsai and E.R. Geertsma for advice on transport experiments and B. Dreier for help with MALS experiments. All members of the Dutzler lab are acknowledged for help in all stages of the project. Data collection was performed at the X06SA and X06DA beamlines at the Swiss Light Source of the Paul Scherrer Institute.

## Author contributions

I.A.E. and J.L. cloned and identified EcoDMT as promising construct in initial screens. I.A.E. and F.M.A. established overexpression and purification and identified initial crystallization conditions. C.M. purified the proteins, improved crystals and determined crystal structures of EcoDMT and mutants. I.A.E. purified the proteins and established transport assays. Transport behaviour was characterized by I.A.E. and C.M. R.D. assisted in X-ray crystallography and structure determination. I.A.E., C.M. and R.D. jointly planned the experiments analysed the data and wrote the manuscript.

## Additional information

**Competing financial interests:** The authors declare no competing financial interests.

