## [Peer Review File · Nature Communications]

Reviewer #1 (Remarks to the Author)

The authors (Ehrnstorfer et al.) report an interesting new structure of prokaryotic SLC11/NRAMP family protein (EcoDMT) in an outward-facing conformation. They also optimized their liposome assay for functional analyses, which was used in the previous work (ScaDMT ; Ehrnstorfer et al., 2014). Based on the EcoDMT structure and liposome assay, they showed that EcoDMT is a proton-coupled metal ion transporter. Together with their previous ScaDMT structure, the authors also provided the detail mechanism of conformational transition in SLC11 transporters, which is important advance in the research field of SLC11 transporters (or broadly other LeuT fold proteins). Since their experiment and discussion are enough and convincing, I think additional experiments are not required. Overall, I think that, if their findings and discussion become clear in the revised manuscript, this paper is worth to be published in Nature communications.

Major points

1) Improvement of figure depictions is needed.

The figures are not clear. Especially, the 3D structure depictions are confusing for readers. Therefore, I highly recommend that the authors should improve figure depictions for clearer understanding. For example, using ball stick model, rotated and close view, or cylinder representation is effective. I also think the stereo view is confusing. In particular, I recommend they should reconsider following parts.

- In Fig 2b, relationships of helices between Fig 2a and 2b are ambiguous. Enclosing in a square is effective.
- In Fig3a, d, e, Superposition image is hard to recognize at a glance. Trying other depiction ways (e.g. highlighting of residues or helices) are effective.
- In Fig4a, as Mn^{2+} ions are not observed in the EcoDTM structure, the black sphere in the left panel is misleading. Furthermore, the colors of Mn^{2+} and gray mesh are similar, which makes readers confusing.
- In Fig5a,b, the structure views are too magnified, and it's hard to grasp the positions of mentioned residues in the overall structures.
- In Fig 7a, where is the area closed-up in Fig 7b? What is the meaning of black circle with slash line between 1a and 1b?
- In Fig 7b, writing the residue numbers are helpful for understanding.
- In Supplementary Fig 3a-c and Supplementary Fig 5a-c, the stereo views colorings are confusing. Contoured at 1σ is low, and contoured at higher threshold (2σ or 3σ) is helpful for clear understanding. Cutting the slab edge is also effective.

- In Supplementary Fig 6a-c, the stereo views and their colorings are confusing.

2) Careful discussion of transition metal binding site is needed.

Based on the sequence homologies, the authors concluded that conserved residues at the unwound region are the transition-metal ion-binding sites (Page 7, line 145). However, it's an overstatement because the species used structure determinations are different and their sequence homology is not so high (~55 %). Therefore, the authors should carefully discuss the transition-metal ion-binding sites. Furthermore, as not all readers know their previously-determined ScaDMT structure, the authors should describe the orientations of ScaDMT structure in more detail. In particular, the description of metal-binding site orientations in ScaDMT structure is helpful for the clear description and discussion of the transition-metal ion-binding sites.

3) Discussion part is too long.

Some of the argument in Discussion part is redundant. Authors should make overlapped explanations cut or transferred to Result part. Clear and concise explanation is essential for readers.

Minor points

1) (Page 3, line 31) "natural resistance-associated protein (NRAMP) family" is misspelling of "natural resistance associated macrophage protein", isn't it?

2) (Page 4, line 74) What is the function of TM12, which are observed only in EcoDMT and human DMT, and not in ScaDMT? More can be mentioned about this helix, based on the EcoDMT structure?

3) (Page 5, line 79) How about showing SEC peak of EcoDMT as similar manner in "Supplementary Figure 2 in ScaDMT paper"?

4) (Pages 7-8, lines 153-160) Writing " α -helix 4" and "C-terminal part of α 4" is confusing. Unified explanation about helix is effective.

5) (Page 14, line 312) Why do authors say "data not shown"? They at least should mention the calculation method or cite the used program (e.g. APBS).

6) (Supplementary Table1) Adding "CC1/2" and "Ramachandran plot (%)" is effective especially for structural biologist.

Reviewer #2 (Remarks to the Author)

The publication Structural and mechanistic basis of proton-coupled metal ion transport in the SLC11/NRAMP family by Dutzler and colleagues describes the high resolution structure of a

secondary transporter using the alternating access mechanism to transport transition metal ions from outside the cell into the cytoplasm. The here presented structure of EcoDMT in an outward facing state elucidates for the first time this exact transport mechanism by merging the data with the inward facing state of ScaDMT, another family member of these transporters, which structure was solved by the same group.

First of all the manuscript is written in a very neat and concise manner, consecutively built up and answering all questions in a logical and clearly understandable manner. The figures are well presented and figure legends are easy to follow with all necessary information. Mutational studies showed in addition why for instance Mn can be still transported but H cannot. The crystallography was performed up to normal standards with maybe a little bit of overrefinement and statistics were absolutely in the range of allowance. The functional data nicely confirms and complements all structural findings.

The research itself is absolutely timely, of highest quality and the combination of structural and functional data gives a holistic understanding of in depth details of this class of transporters, of which all existing structures are solely solved by the Dutzler group. These transporters are of very high medical importance too, as homologues are reported to be involved in a wide range of resistances against drugs.

In summary I would strongly recommend to publish this manuscript without any changes.

Reviewer #3 (Remarks to the Author)

This is a nice paper describing very technically challenging work which further defines the mechanism by which SLC11 transporters mediate metal ion/proton cotransport. The authors have solved the structure for a prokaryotic family member and have compared that structure to those of other family members, including the human DMT1 cotransporter. While the findings appear to be sound and the conclusions properly drawn from the presented data, the overall significance of this work is unclear. The priority of this work is lessened somewhat since mechanisms/structures of other family members have previously been worked out. Moreover, there is no clear articulation by the authors of the importance of this work in a practical sense. These perceptions decrease the overall enthusiasm for this investigation.

Response to reviewers' comments:

Reviewer #1 (Remarks to the Author):

The authors (Ehrnstorfer et al.,) report an interesting new structure of prokaryotic SLC11/NRAMP family protein (EcoDMT) in an outward-facing conformation. They also optimized their liposome assay for functional analyses, which was used in the previous work (ScaDMT ; Ehrnstorfer et al., 2014). Based on the EcoDMT structure and liposome assay, they showed that EcoDMT is a proton-coupled metal ion transporter. Together with their previous ScaDMT structure, the authors also provided the detail mechanism of conformational transition in SLC11 transporters, which is important advance in the research field of SLC11 transporters (or broadly other LeuT fold proteins). Since their experiment and discussion are enough and convincing, I think additional experiments are not required. Overall, I think that, if their findings and discussion become clear in the revised manuscript, this paper is worth to be published in Nature communications.

We thank the reviewer for the positive comments.

Major points

1) Improvement of figure depictions is needed. The figures are not clear. Especially, the 3D structure depictions are confusing for readers. Therefore, I highly recommend that the authors should improve figure depictions for clearer understanding. For example, using ball stick model, rotated and close view, or cylinder representation is effective. I also think the stereo view is confusing.

In our revised version, we have replaced several stereo figures, (e.g. Fig 2a, Fig S3a and b, Fig S5a and b) but we have kept others where we thought that stereo was essential for a full comprehension of the structure. The included stereo figures are of very high quality and, if printed at the right size (at total width of 11 cm), they are straightforward to view. I am sure that readers who make the effort will find them very useful. We have used similar representations for over 20 years in all of our manuscripts. They are essential to illustrate complex structural relationships, particularly in all atom representations or superpositions. In our past publications reviewers usually requested the inclusion of additional stereo figures.

In particular, I recommend they should reconsider following parts.

- In Fig 2b, relationships of helices between Fig 2a and 2b are ambiguous. Enclosing in a square is effective.

We have now removed the stereo panel of Fig. 2a and present the structure as cylinder representation in two different orientations with selected helices labeled. In the orientation that is equivalent to Fig. 2b (right) we have indicated the region shown in Fig 2b as a box.

- In Fig3a, d, e, Superposition image is hard to recognize at a glance. Trying other depiction ways (e.g. highlighting of residues or helices) are effective.

We decided to keep Figure 3a as stereo figure but we have labeled selected helices to facilitate orientation. The figure shows a superposition of two structures that is the basis for the quantification of conformational differences shown in Fig. 3b. This superposition can only be accurately represented as C α -trace since cylinder models provide an approximation of the secondary structure that does not allow for an accurate comparison of conformational differences. Due to the complexity of the image, a stereo view is in this case essential. Whereas Figure 3a provides an overview of the entire structure, Figures 3d and e show relevant sections of the structures that illustrate the regions of largest movements. We have improved both panels to make them easier to comprehend. Figure 3d now provides a view from the cytoplasm and Figure 3e from the outside. Both figures now only show helices that change their conformation and they contain appropriate labels for inward and outward facing conformations.

- In Fig4a, as Mn²⁺ ions are not observed in the EcoDTM structure, the black sphere in the left panel is misleading. Furthermore, the colors of Mn²⁺ and gray mesh are similar, which makes readers confusing.

To better distinguish them from the molecular surface, we now show the Mn²⁺ ions as green spheres in all of our figures. We have also removed them from the representation of EcoDMT in our main Figures (Figs. 4a and Fig. 5a and b) where we have shown them previously to facilitate the orientation. In all cases where we still show a Mn²⁺ ion bound to EcoDMT in the Supplementary materials (Supplementary Fig. 4b and e, Supplementary Fig. 7a and c) we state in the legend that its position was modeled.

- In Fig5a,b, the structure views are too magnified, and it's hard to grasp the positions of mentioned residues in the overall structures.

In our revised figure we have now included the helices containing the respective residues.

- In Fig 7a, where is the area closed-up in Fig 7b? What is the meaning of black circle with slash line between 1a and 1b?

The white and black circles have indicated the location of hinge regions that approximate the rigid body movements of helices during the exchange of conformations. We have now replaced them by circles and clarified their meaning in the figure legend. Fig. 7b provides a schematic view of the ion binding site that changes its access from an outward to an inward-facing conformation including a possible H⁺ pathway that is displayed Fig 5b in its structural detail. The aqueous cavities pointing either to the outside or inside are indicated in both panels in light blue which should facilitate the orientation.

- In Fig 7b, writing the residue numbers are helpful for understanding.

The detailed residue numbers are displayed in Fig. 5b. Their inclusion in Fig. 7b would make the figure very crowded.

- In Supplementary Fig 3a-c and Supplementary Fig 5a-c, the stereo views colorings are confusing. Contoured at 1σ is low, and contoured at higher threshold (2σ or 3σ) is helpful for clear understanding. Cutting the slab edge is also effective.

The representation of $2F_o-F_c$ density at 1σ is custom and fully appropriate, particular in this case, where due to the high Wilson B-factor the map had to be sharpened to provide a detailed representation of the side-chain density. A higher contour (3σ) is usually chosen in case of difference densities. In our revised Supplementary Figs. 3a and b we have replaced the stereo figures with figures showing electron density around α -helices 1 and 6. Similarly we have removed stereo views in Supplementary Figs. 5a-c and now present the regions around the introduced mutation as single panels. We have kept the stereo representation for Supplementary Fig. 3c since we still think that it provides a more instructive overview of the quality of the electron density. In our revised manuscript we have now also included the anomalous difference densities at the site of mutations of four novel methionine mutants.

- In Supplementary Fig 6a-c, the stereo views and their colorings are confusing.

Supplementary Fig. S6a shows the superposition of different conformations of LeuT, Supplementary Figs. S6b and c superpositions of the two SLC11 structures with their corresponding conformations in LeuT. As stated before, such superpositions are complex and thus require stereo representations. The coloring was chosen to have unique colors for the N and C-terminal halves of both transporters, and, in case of the superposition of different conformations of the same protein (such as in Fig. 3a or Supplementary Fig. 6a), darker and brighter versions of the same color. We have clarified this in the respective figure legends.

2) Careful discussion of transition metal binding site is needed. Based on the sequence homologies, the authors concluded that conserved residues at the unwound region are the transition-metal ion-binding sites (Page 7, line 145). However, it's an overstatement because the species used structure determinations are different and their sequence homology is not so high (~55 %).

We disagree that the similarity between both prokaryotic transporters would be insufficient to conclude on the conservation of the site. With a sequence identity of 53% and similarity of 67% the conservation between both proteins is exceptionally high. In our previous work on different ion channel and transport proteins, we found functional equivalence of important residues in family members with much lower sequence similarity. In our previous work, we have demonstrated the functional equivalence of the ion-binding site residues between ScaDMT and human DMT1 by mutagenesis and we show a similar relationship for EcoDMT in this paper (see Fig. 4b and c).

Therefore, the authors should carefully discuss the transition-metal ion-binding sites. Furthermore, as not all readers know their previously-determined ScaDMT structure, the authors should describe the orientations of ScaDMT structure in more detail. In particular, the description of metal-binding site orientations in ScaDMT structure is helpful for the clear description and discussion of the transition-metal ion-binding sites.

We have added an additional panel (Supplementary Fig. 4a) to show the transition metal ion binding site of ScaDMT and introduced a more detailed description of this site in the results:

Page 7:

Helices 1 and 6 are both unwound in the center of the lipid bilayer and, as revealed in the ScaDMT structure¹⁶, conserved residues of this unwound region constitute the transition-metal ion-binding site (Fig. 2b). In this site, ion-protein interactions are established with a backbone carbonyl at the end of α -helix 6a, and the side chains of an aspartate and an asparagine, all of which act as hard ligands that would also be suitable for the coordination of Ca^{2+} . In contrast, the side chain of a methionine that is also part of the binding site acts as transition metal specific soft ligand that would not be suitable for Ca^{2+} coordination^{16,22} (Supplementary Fig. 4a).

3) Discussion part is too long. Some of the argument in Discussion part is redundant. Authors should make overlapped explanations cut or transferred to Result part. Clear and concise explanation is essential for readers.

We have shortened the discussion at several places. The remaining points are essential to put the results of this work into a broader context.

Minor points

1) (Page 3, line 31) "natural resistance-associated protein (NRAMP) family" is misspelling of "natural resistance associated macrophage protein", isn't it?

We have corrected the mistake.

2) (Page 4, line 74) What is the function of TM12, which are observed only in EcoDMT and human DMT, and not in ScaDMT? More can be mentioned about this helix, based on the EcoDMT structure?

We think that α -helix 12, which is located at the periphery of the protein, does not play a central role during transport. This is compatible with the role of the equivalent helix in other transporters sharing the same fold and also accounts for the fact that the helix is not present in several prokaryotic transporter (such as ScaDMT).

We have introduced a sentence in the result section:

The last helix of EcoDMT, which is absent in ScaDMT, is located at the periphery of the protein and does probably not play a major role during transport (Fig. 2a).

3) (Page 5, line 79) How about showing SEC peak of EcoDMT as similar manner in "Supplementary Figure 2 in ScaDMT paper"?

We have included a SEC-MALS experiment as novel panel to Supplementary Fig. 2.

4) (Pages 7-8, lines 153-160) Writing " α -helix 4" and "C-terminal part of $\alpha 4$ " is confusing. Unified explanation about helix is effective.

We have corrected the helix nomenclature.

5) (Page 14, line 312) Why do authors say "data not shown"? They at least should mention the calculation method or cite the used program (e.g. APBS).

The reference to the continuum electrostatics calculations was removed in the revised version of our manuscript. We have carried out these calculations with the PBEQ module of the simulation program CHARMM. Although we found that ionizable residues had predicted pK shifts in a meaningful direction (i.e. towards higher values for E129 and lower values for H236), the magnitude of the shifts were for some residues unrealistic. We thus decided not to include these calculations in our study.

6) (Supplementary Table1) Adding "CC1/2" and "Ramachandran plot (%)" is effective especially for structural biologist.

We have included the CC1/2 values in Table 1 and Supplementary Table 1 and added the Ramachandran statistics in the methods.

Reviewer #2 (Remarks to the Author):

The publication Structural and mechanistic basis of proton-coupled metal ion transport in the SLC11/NRAMP family by Dutzler and colleagues describes the high resolution structure of a secondary transporter using the alternating access mechanism to transport transition metal ions from outside the cell into the cytoplasm. The here presented structure of EcoDMT in an outward facing state elucidates for the first time this exact transport mechanism by merging the data with the inward facing state of ScaDMT, another family member of these transporters, which structure was solved by the same group. First of all the manuscript is written in a very neat and concise manner, consecutively built up and answering all questions in a logical and clearly understandable manner. The figures are well presented and figure legends are easy to follow with all necessary information. Mutational studies showed in addition why for instance Mn can be still

transported but H cannot. The crystallography was performed up to normal standards with maybe a little ebit of overrefinement and statistics were absolutely in the range of allowance. The functional data nicely confirms and complements all structural findings. The research itself is absolutely timely, of highest quality and the combination of structural and functional data gives a holistic understanding of in depth details of this class of transporters, of which all existing structures are solely solved by the Dutzler group. These transporters are of very high medical importance too, as homologues are reported to be involved in a wide range of resistances against drugs. In summary I would strongly recommend to publish this manuscript without any changes.

We thank the reviewer for these supportive remarks. With respect to refinement we can assure that we were cautious not to overfit the model during the refinement process. In case of EcoDMT this was a challenging task since, due to the lack of non-crystallographic symmetry (NCS), we were unable to improve the initial phases by NCS symmetry averaging, which has complicated the interpretation of loop regions during model building. Later, the lack of structural redundancy in the asymmetric unit has resulted in an unfavorable observable to parameter ratio that decreased the radius of convergence during refinement. The small separation of R and R_{free} of only 3.3% ensures that the structure was not overrefined. It should be emphasized that R_{free} in case of NCS is not fully independent and the separation of R to R_{free} in case of NCS symmetry is thus generally lower. We also want to point out that we have made large efforts to improve our structure for this revision and that we have extended the model at the previously not interpreted N-terminus by 26 residues. For that purpose, we have constructed four additional methionine mutants and collected anomalous data from crystals of Se-Met derivatized protein to confirm the register of this region (Supplementary Fig. 3e). This extension has further improved the refinement statistics (Table1).

Reviewer #3 (Remarks to the Author):

This is a nice paper describing very technically challenging work which further defines the mechanism by which SLC11 transporters mediate metal ion/proton cotransport. The authors have solved the structure for a prokaryotic family member and have compared that structure to those of other family members, including the human DMT1 cotransporter. While the findings appear to be sound and the conclusions properly drawn from the presented data, the overall significance of this work is unclear. The priority of this work is lessened somewhat since mechanisms/structures of other family members have previously been worked out. Moreover, there is no clear articulation by the authors of the importance of this work in a practical sense. These perceptions decrease the overall enthusiasm for this investigation.

We thank the reviewer for these remarks. Whereas there are now several structures of transporters with a similar fold in different conformations, the SLC11 family distinguishes itself through two important features. While distantly related transporters carry amino acids, sugars or small organic molecules, members of the SLC11 family constitute one of the most

important transport systems for transition metal ions. This requires unique mechanisms for the recognition of substrates, which we have elucidated in this and a previous study. In addition, the coupling to H^+ instead of Na^+ is a unique feature of the SLC11 family that increases its versatility to either transport metal ions across the plasma membrane or out of endosomes. The coupling to protons is a property that is still poorly understood but that is central to many intracellular transport processes. We thus believe that our work is of interest for a broader audience beyond the group of scientists interested in transition metal ion transport.

Besides pointing out the relevance of the family in the introduction, we have thus added the following sentences to the discussion:

Page 13:

The coupling to H^+ provides a source of energy for active transport in such different environments as the plasma membrane and intracellular organelles. Despite of their abundance in pro- and eukaryotic organisms, the mechanisms of proton-coupled transporters are to date still poorly understood.

We also have introduced references to previous studies that have shown that a mutation of a conserved residue of the putative proton exit path in DMT1 was mutated in an anemic patient.

Page 15:

Interestingly, Arg416 in human DMT1, which corresponds to Arg373 in EcoDMT, was found to be mutated in a patient with severe anemia^{42,43}

Reviewer #1 (Remarks to the Author)

I am satisfied that the authors have made several critical changes to the manuscript to address the concerns that I raised. Especially, figures are much clearer than previous version. Therefore, I think that the author's manuscript is now acceptable for publication.

Reviewer #3 (Remarks to the Author)

The revisions are acceptable.